# Structures of the ß-Keratin Filaments and Keratin Intermediate Filaments in the Epidermal Appendages of Birds and Reptiles (Sauropsids)

**DOI:** 10.3390/genes12040591

**Published:** 2021-04-17

**Authors:** David A.D. Parry

**Affiliations:** School of Fundamental Sciences, Massey University, Private Bag 11-222, Palmerston North 4442, New Zealand; d.parry@massey.ac.nz; Tel.: +64-6-9517620; Fax: +64-6-3557953

**Keywords:** corneous ß-proteins, keratin intermediate filaments, X-ray fiber diffraction

## Abstract

The epidermal appendages of birds and reptiles (the sauropsids) include claws, scales, and feathers. Each has specialized physical properties that facilitate movement, thermal insulation, defence mechanisms, and/or the catching of prey. The mechanical attributes of each of these appendages originate from its fibril-matrix texture, where the two filamentous structures present, i.e., the corneous ß-proteins (CBP or ß-keratins) that form 3.4 nm diameter filaments and the α-fibrous molecules that form the 7–10 nm diameter keratin intermediate filaments (KIF), provide much of the required tensile properties. The matrix, which is composed of the terminal domains of the KIF molecules and the proteins of the epidermal differentiation complex (EDC) (and which include the terminal domains of the CBP), provides the appendages, with their ability to resist compression and torsion. Only by knowing the detailed structures of the individual components and the manner in which they interact with one another will a full understanding be gained of the physical properties of the tissues as a whole. Towards that end, newly-derived aspects of the detailed conformations of the two filamentous structures will be discussed and then placed in the context of former knowledge.

## 1. Introduction

Structural studies on the epidermal appendages of birds and reptiles (sauropsids) presented in this review will draw on data, not only from the birds, but also from the crocodiles, turtles, lizards, snakes, and tuatara. An appreciation of the relationships between these vertebrates is, therefore, of some value and towards that end the phylogenetic classification of the sauropsids is shown in Figure 1. This illustrates that the archelosaurs diverged into the archosaurs (birds and crocodiles) and the testudines (turtles) about 272 Mya, and that the lepidosaurs diverged in to the squamates (lizards and snakes) and the rhynchocephalia (tuatara) about 230 Mya [1,2,3,4]. Whilst this diversification will necessarily result in important differences between the species it is expected that major similarities will remain, a supposition borne out by much of the data presented here.

Electron microscopy investigations of the cross-sections of the hard keratins constituting the epidermal appendages (feathers, claws, scales) of the sauropsids have shown that these tissues have a filament-matrix texture [6,7,8,9,10,11,12]. This is dominated by filaments about 3.4 nm in diameter that provide much of the tensile strength of the tissue (Figure 2). The matrix formed by the terminal domains of the filament-forming corneous ß-proteins (CBP) and the other proteins of the epidermal differentiation complex (EDC) [13], in contrast, has an important role in specifying hardness, toughness, pliability, compression attributes, and resistance to torsion and flexibility [14,15,16,17,18,19,20,21,22]. A second type of filament, about 7–10 nm in diameter, has also been observed in these appendages. These filaments—the keratin intermediate filaments (KIF)—are less common than their smaller counterparts [23] and tend to occur in discrete layers akin to those sometimes seen in the mammalian hard keratins, such as hair and quill [24,25,26,27,28,29,30]. Their terminal domains, likewise, form an important part of the matrix structure.

Over the past 20 years, hundreds of sequences of closely related families of corneous ß-proteins (CBP, also known as ß-keratins; [32,33]) have been published. These sequences represent every branch in the phylogenetic classification of the birds and reptiles and include, for example, those from chicken and zebra finch [34], crocodile and alligator [28,35], turtle [29], tuatara [33,36], green anole lizard [37], and snake [26]. Chains from each of these protein families form the 3.4 nm diameter filaments that exist in that particular species. A much smaller number of sequences of the closely related families of chains that assemble to form the 7–10 nm KIF in the sauropsids have been published thus far (*Anolis carolinensis*, [38,39,40]; *Sphenodon punctatus*, [36]). Both sets of sequence data (CBP and KIF) have provided new insights into the structures of the two filament types, and these will be described in Section 2 and Section 3, respectively.

X-ray diffraction studies on the 3.4 nm filaments and their constituent CBPs have largely been confined to those based on seagull feather rachis (*Larus novae-hollandiae*). This material in its natural form contains highly orientated filament bundles, thereby making the rachis ideally suited to provide the most detailed X-ray patterns possible (Figure 3a). The well-defined row lines present that indicate lateral order (Figure 3c) can be removed by pressing the specimen in steam (Figure 3d). Less detailed X-ray data have also been obtained from other materials, including goanna claw (Figure 3e; *Varanus varius*, [24])*,* chicken scale (*Gallus gallus domesticus*, [41]), snake scale (*unidentified species*, [24]) and tuatara claw [42].

Structural studies of KIF using fibre X-ray diffraction also rely to a very high degree on the best possible molecular orientation existing in the specimen used. To date, this has precluded the use of KIF isolated from the sauropsids. However, considerable detailed structural studies have been carried out on KIF from mammalian α-keratin quill in which the molecular orientation is naturally high (Figure 3b). Importantly, recent sequence studies have revealed that the KIF proteins from the sauropsids display substantial homology to those from the mammalian hard α-keratins [36,38]. It follows that the molecular structures of the KIF chains and their arrangement in the filaments will be very similar in the sauropsids and in the mammals and, indeed, in all probability many or most features will be identical. The general conclusions of mammalian KIF research is, therefore, highlighted here for the first time in a review on the structures of the sauropsid epidermal appendages.

## 2. The 3.4 nm Diameter Filaments Formed from the Corneous ß-Proteins

Many techniques have been employed in an effort to gain a deeper understanding of the structures of the constituent CBP molecules, their assembly into filaments and the lateral assembly of those filaments with respect to one another. Such techniques include electron microscopy, fiber X-ray diffraction, infrared spectroscopy, protein chemistry, sequences analyses, and model building. The 3.4 nm diameter filaments are a major constituent of all of the epidermal appendages present in the sauropsids and, as such, it is important to understand their detailed structures if we are to understand their mechanical attributes. It is noted, of course, that although these filaments are but one component in the tissue as a whole, their contribution is clearly a major one and well worthy of detailed study. This review has therefore described the structure of the molecules and their assembly into filaments in Section 2.1, the lateral organization of the filaments in Section 2.2, and the physical properties of the appendages as a whole in terms of sequence features in Section 2.3.

### 2.1. Structure of the Corneous ß-Protein Molecules and Their Assembly into Filaments 

Early X-ray diffraction patterns using specimens of well-orientated feather rachis revealed a highly regular structure that was believed to originate from the filamentous component of the rachis (Figure 3a). Since this was generally similar to the pattern obtained from stretched α-keratin, it was suggested that the conformation of the molecules in the 3.4 nm diameter filaments of avian and reptilian keratins was based on the ß-sheet conformation, rather than that of the α-helix [45]. More detailed research showed that the filaments consisted of a helical arrangement of small ß-crystallites with four-fold screw symmetry [46,47,48]. The pitch length (*P*) of the helix in feather keratin was 9.6 nm and the axial rise (*h*) per repeating unit was 2.4 nm. Small changes in *P* have been observed in some tissues (9.2 nm in chicken scale [41]; 9.28 nm in snake scale [24]; 9.85 nm in lizard claw [24]) but the four-fold screw symmetry is maintained. It would seem probable, therefore, that the basic helical framework of the filaments is conserved across all of the sauropsids (Figure 3d–e).

The first sequence determined for any CBP was that from emu feather [49]. One enzymatically-derived fragment was shown by infrared spectroscopy to have a high ß-sheet content with the chain looped back and forth in an antiparallel conformation [50,51]. A subsequent analysis of the whole sequence revealed that the ß- and turn-favoring residues were confined to a central portion of the chain and that these residue groupings had out-of-phase periodicities of about eight residues [52] (Figure 4). The predicted length of the ß-strands was thus about 2.4 nm, a value that corresponded precisely to the meridional repeat derived from the X-ray diffraction patterns. Sequence analyses by Sawyer et al. [53] recognized the high degree of homology across the avian CBPs and also noted that a 20 residue segment existed in the bird sequences that was homologous to that in alligator claw. This same region was seen in the archosaurs and squamates [54] and it was, therefore, suggested [37] that this region played a role in assembly. A sequence homologous to that recognized earlier in emu feather with the postulated ß-conformation was subsequently observed in a reptilian claw [55,56], and since that time it has been shown that a ß-containing central domain 34-residue long that encompasses the former 20 residue region is highly conserved in the CBPs across all species and appendages [22,31,57,58]. Imediately N-and C-terminal to the 34-residue sequence there are short pieces of sequence about 7–11 residues long that are also largely conserved and it has been speculated that 142 these are likely to be involved in some way in filament assembly [22]. From model building, it can be deduced that the ß-crystallites will form the framework of the filaments. Using the quantitative fit to the X-ray diffraction patterns as a criterion for the model building it was further shown that the crystallites must be formed from a pair of antiparallel ß-sheets—a ß-sandwich [59,60]—and that these sheets will be related to one another by a perpendicular dyad axis of rotation. Further, the X-ray data demanded that the sheets could not be planar but must be twisted in what is now believed to be a right-handed manner [61].

A number of other, key observations provided support for these conclusions. Firstly, one face of the ß-sheet in the 34-residue repeat was shown to be largely composed of apolar residues (amphipathic), thereby providing a ready and simple means by which a pair of these sheets could assemble via their apolar faces to form a ß-sandwich [30]. The amphipathic nature of one face of the ß-sheet was independently confirmed [62] using a molecular mechanics and Poisson Boltzmann approach. Secondly, the sequences of the central turns 2 and 3 in the ß-crystallites (consensus sequences Q/R-P-P/S-P and L/I-P-G-P, respectively) were strongly conserved across all chains implying that the axial assembly of the ß-crystallites will be dependent on the strong apolar interactions that these residues would provide (Figure 5; [22,57,62]). 

Modeling of the ß-sandwich with a perpendicular dyad axis relating the two constituent ß-sheets indicates that the latter will contain three central ß-strands and two partial outer ones. This, naturally, leads to close packing between the sheets of the residues in the inner strands and occurs in layers perpendicular to the fiber axis [57]. In an alternative model [62], each ß-sheet would contain four central ß-strands and the two sheets would be related to one another by a parallel dyad axis of rotation. However, close packing of residues between the sheets would be compromised and, furthermore, parts of the X-ray diffraction pattern (even layer lines in the 0.3 nm region and odd layer lines in the 0.6 nm region) would be less compatible with such a model [61,63].

The terminal domains of the CBP, i.e., those sequences immediately N- and C-terminal to the conserved 34-residue region show considerable substructure but one that differs between the archelosaurs and the lepidosaurs (Figure 4; [31]). In the former case (birds, crocodiles, and turtles), the N-terminal domain is comprised of a single domain (subdomain A), whereas in the latter (tuatara, snakes, and lizards) there are two subdomains, A and B. Subdomain A is typically about 25 residue long and is cysteine- and proline-rich, and subdomain B is variable in length (20–76 residues) and contains multiple sequence repeats based on glycine, serine, alanine, leucine, and the aromatic residues. In tuatara, three types of N-terminal domains have been recognized but with this exception the pattern discussed above is maintained [33]. The C-terminal domain of all the sauropsids comprises two subdomains, C and D. Subdomain C, like subdomain B, is variable in length (20–128 residues) but, unlike subdomain B, is essentially rich in just glycine and tyrosine residues, often in repeating sequences. Subdomain D is short (9–18 residues), cysteine-, arginine- and lysine-rich, and occurs at the C-terminal end of the chain. It will be indicated in Section 2.3 that the terminal domains are important in specifying the physical attributes of the epidermal appendages and that an appropriate mix of family members might have the capability of fine-tuning these properties.

### 2.2. Lateral Arrangement of the 3.4 nm Diameter Filaments

A feature of the lepidosaurs that is not shared with the archelosaurs is the presence of a CBP chain with four 34-residue repeats. This has a higher molecular weight (33–41 kDa) than the majority of the CBP chains (10–18 kDa) and has been observed in the king cobra and Burmese python [64], Japanese gecko [65] and green anole lizard [37], and in New Zealand tuatara (Figure 6; [35]). In theory, such a chain is capable of initiating filament assembly [5] or forming interfilamentous linkages to provide lateral reinforcement to the epidermal appendage [63]. 

An observation [66] has relevance for the first possibility. They showed that the disulphide bonds in seagull feather rachis could be reduced and substituted to give a soluble protein. Given the appropriate conditions, the protein was then shown to be capable of re-assembly into filaments that, in the electron microscope, appeared indistinguishable from native ones. Further, fibers of reconstituted filaments used as specimens in X-ray diffraction experiments revealed a characteristic 2.4 nm meridional reflection, as seen in native rachis. Thus, no co-factors are required for filament assembly to proceed in vitro and the protein is capable of self-assembly through apolar interactions between the faces of a pair of ß-sheets, allied to shape complementarity, to form a ß-sandwich. In turn, these ß-sandwiches are capable of axial assembly to form an intact filament. The consequence of this result is that a CBP chain with four 34-residue repeats is not a requisite for filament formation though it might add stability to the initial stages of assembly in vivo.

In the case of potential lateral linkages, it was shown that six families of structures spanning two, three, or four filaments were theoretically possible [63]. Subsequent modeling [67] indicated, however, that one particular model had particular merit. This involved a ß-sandwich in one filament being connected to a ß-sandwich in a different filament, but with both ß-sandwiches existing within the same four 34-residue repeat chain (Figure 7). Such linkages would exist only between two neighboring filaments but a random distribution of them would provide important lateral reinforcement to the tissue as a whole. More experimental work will be required to confirm this conclusion but the concept has clear physical implications.

A characteristic series of row lines on a 3.4 nm lattice has been observed in the fiber X-ray diffraction patterns of feather keratin but, so far, only feather keratin. Whether or not better oriented specimens from other sauropsids would reveal a similar feature in their X-ray patterns remains an open question. The interpretation of this observation is that the side-to-side packing of filaments is orthogonal and that the filaments are lined up equivalently in a direction perpendicular to the fiber axis [46,68]. This is consistent with the electron microscope observations [6], which recognized small sheets containing 4–10 filaments. These sheets were relatively straight but were nonetheless sufficiently pliant to permit some small distortion to occur. Fraser and Parry [44] systematically varied the orientation of the filaments within these sheets and compared the calculated diffraction pattern with that observed. They were able to achieve a good agreement when the filaments were all oriented in such a way that the perpendicular dyad axis was inclined at an angle of 17° with respect to the plane of the sheet. Interestingly, this resulted in the ß-sheets in neighboring filaments lining up (Figure 8). While this observation was not interpretable at the time the postulate that the four 34-residue chains might link neighboring filaments via their central linker domains, which are thought to exist in a ß-like conformation, does provide a rationale for the earlier observation, though further proof will be required [67].

### 2.3. Physical Properties of the 3.4 nm Diameter Filaments in Terms of Sequence Features

The physical properties of the epidermal appendages are dependent on many factors. These would likely include the structures of the two filamentous components, as well as the matrix of proteins that surround them, and the interactions that occur between the filaments and the matrix. The latter, unfortunately, are unknown at the present time thus restricting any analysis, such as this, to ascertaining the properties that individual components might bring to the tissue as a whole. The fact that it is families of closely related proteins that make up both sets of filaments implies that the matrix, formed by the terminal domains, can be varied by incorporating the “right” mixture of chains to fine tune the mechanical properties that are required for the appendages to function optimally. The subdomain structure noted earlier is likely to play an important part [22,58]. In essence, the subdomains may be classified as either highly charged cysteine-rich, glycine-tyrosine-rich, or glycine-rich.

The charged residue component of the highly charged cysteine-rich subdomains will necessarily be located on the surface of a protein, and, thus, be in a position where the charged residues may be hydrated. Studies by Taylor et al. [21] on the effect of hydration on the tensile properties of avian keratins revealed that an increase in water content renders it more pliable and generally softens the tissue. The cysteine residues, on the other hand, will result in the formation of more disulphide bonds, thereby ensuring that the tissue remains insoluble and that its resistance to proteolysis is not compromised.

The glycine-tyrosine-rich subdomains (also known as the HGT proteins in the context of the keratin-associated-proteins in mammalian α-keratins) are of especial interest. Glycine and tyrosine residues commonly occur in the interior of a protein structure. Each residue type brings a unique attribute to these subdomains. Glycine has the smallest sidechain of any residue, it is apolar and it permits the greatest possible conformational freedom for the chain in terms of the wide range of possible rotations that may occur about its mainchain single bonds. Tyrosine is much larger than glycine, it too is apolar and, most importantly, it has an aromatic ring and a reactive hydroxyl group. The tyrosine residues would, therefore, be expected to interact strongly with one another through ring stacking or through the formation of multiple hydrogen bonds [22,58]. The chain flexibility resulting from the many glycine residues present would greatly facilitate the likelihood of both forms of interactions occurring. These high-glycine-tyrosine features would be expected to strengthen the material, while also permitting limited pliability.

Glycine-rich subdomains are a major feature, not only of the corneous ß-proteins, but also of the terminal domains in the epidermal and epithelial keratins of mammals. In that context, it was suggested by Steinert et al. [69] that these would adopt a series of glycine loops anchored on interactions between large apolar residues aperiodically disposed along the sequences. Because glycine loops are strongly apolar, it was further postulated that they would be able to adopt a wide variety of energetically-similar, compact conformations that would largely shield the glycine residues from the aqueous environment. Such a family of conformational states would be compatible with an imperfectly organized matrix and would provide a degree of elasticity to the tissue. There is every reason to believe that a similar interpretation of the glycine-rich domains is pertinent to the corneous ß-proteins.

## 3. Structure of the Keratin Intermediate Filament (KIF) Molecules and the 7–10 nm Diameter Filaments

Much of the structural research on intermediate filaments has been carried out using specimens of trichocyte keratin, such as quill and hair, primarily because the degree of orientation of the IF present in these specimens is naturally high. Quill, in particular, has allowed excellent X-ray diffraction patterns to be obtained (Figure 3b), and these have been sufficiently detailed to allow considerable conformational information to be ascertained. Transmission electron microscopy has, likewise, benefitted from the use of well-orientated specimens (Figure 9). The other members of the same family of trichocyte keratins (wool, hoof, horn, claw, and baleen) have more frequently been used in the context of protein chemical analyses and mechanical studies. 

This review on the 7–10 nm diameter filaments in sauropsids will commence with an analysis of the sequences of the KIF proteins from which they are formed (Section 3.1). A comparison will be made with their mammalian counterparts and this will demonstrate that a structural study of one will reveal relevant information about the other. Section 3.2 will concentrate on the molecular conformation of the KIF proteins, as determined by fibre X-ray diffraction studies, and will also describe the surface lattice structure on which the molecular repeating units lie in each of the two structural states adopted by KIF in vivo. Since there are indications that the sauropsids do not contain the matrix proteins or keratin-associated proteins (KAP) that are found in mammalian trichocyte keratins, only a brief discussion will be given of the evidence that supports this conclusion (Section 3.3).

### 3.1. Sequence of KIF Chains in Sauropsids 

From genomic studies it has been shown that families of KIF protein sequences are present in both lizard (40 chains: [38,40]) and tuatara (25 chains from the partial genome: [36]). In addition, similar sequences of KIF proteins in birds [72] and in various terrestrial vertebrates [73] have been reported, and these have also added significantly to the database of sequences now available. A comparison of these sequences with those from mammals and, indeed, from vertebrates in general, reveals that the chains in each species fall naturally into the Type I and Type II classes as initially defined for sheep wool keratin [74,75]. This feature is present in all vertebrate epidermal appendages (see, for example, [71,76,77]). Each chain follows a common sequence pattern with a long, central, heptad-containing region comprised of segments 1A, 1B and 2 with connecting linkers L1 and L12, enclosed by (generally) non-helical N- (head) and C- (tail) domains (Figure 10). The segments displaying a heptad substructure, which are of the form (*a-b-c-d-e-f-g*)_n_, where *a* and *d* are generally apolar residues, are highly conserved in length (segment 1A—35 residues; segment 1B—101 residues; segment 2—148 residues), as is the length of linker L12 (16 residues for all Type I chains and 17 residues for all Type II chains). Linker L1 is a variable length linker. In addition, there are small but significant differences in the sequences in the heptad-containing and linker domains that are characteristic of either the Type I or the Type II chains (see the consensus sequences of both Type I and Type II chains, [78]). The head and tail domains also contain short regions known as H1 and H2; these lie immediately adjacent to the N-terminal end of segment 1A and the C-terminal end of segment 2, respectively. It was shown by Steinert and Parry [79] that the H1 subdomain in the Type II chain in epidermal keratin from mammals has considerable importance in facilitating aggregation of the molecules at the two- to four-molecule level. It is, therefore, of interest that the sequence of this region is conserved in both lizard and tuatara, as well as in human (for example). 

The amino acid compositions of the head and tail domains of the α-keratins (KIF chains), which are generally subdivided into those that are trichocyte, epidermal, or epithelial in origin, are also very characteristic [36,77]. Vertebrate trichocyte keratins, for example, have cysteine-rich heads (Type I—heads, 9%, tails, 18%; Type II—heads, 10%, tails, 12%), whereas epidermal keratins are extremely rich in glycine residues (Type I—heads, 48%, tails, 57%; Type II—heads, 37%, tails 41%). A detailed study of the head and tail domains in tuatara and human KIF chains has shown considerable similarities between vertebrate and tuatara but also some important differences. The similarities include the glycine contents in the Type II epidermal chains in the two species (head, 38 and 37%, respectively; tail, 38 and 41%, respectively). The differences, however, include the glycine contents in the Type I epidermal chains (head, 41 and 48%, respectively, but, most particularly, in the tail, 10 and 57%, respectively). The cysteine residue contents in the head domain of the Type I trichocyte keratin are the same in tuatara and vertebrates (8 and 9%, respectively) but different in the tails (11 and 18%, respectively), whereas in the Type II trichocyte keratin chains the cysteine residues contents are different in the heads (5 and 10%, respectively) but the very similar in the tails (10 and 12%, respectively). All of the sequence data from lizard and tuatara indicate a high probability that the birds, crocodiles, turtles, and snakes will also contain similar KIF chains. It also follows that KIF formed from these chains are viable in the sauropsids in general, a conclusion supported experimentally by the observation in the electron microscope of filaments 7–10 nm in diameter.

### 3.2. Molecular Structure and Arrangement in the KIF

From Section 3.1, it follows directly that the molecular structure adopted by the KIF chains in the sauropsids is very likely to be identical to that seen in the mammals, and that structural data determined from X-ray diffraction and other methodologies on vertebrate (but non-sauropsid) sources will be strongly indicative of a similar feature being present in the sauropsids. With that in mind, earlier work on mammalian α-keratins will now be discussed. 

The structure of the mammalian α-keratins has been investigated since the 1930s, when early X-ray diffraction experiments showed a number of characteristic features in the so-called α-pattern—0.15 and 0.515 nm meridional reflections, a 0.98 nm equatorial reflection, and a near equatorial maxima. These data are consistent with the heptad-containing sequences forming a two-stranded α-helical coiled coil structure with an average pitch length of about 15 nm [80,81,82]. Crystallographic studies on intermediate filaments from various sources have nonetheless indicated that considerable local variations in pitch length do exist (see, for example, the review [83]). There is also a strong body of evidence that the KIF molecules are heterodimers with both a Type I and a Type II chain [84,85,86,87,88,89], and that the chains are parallel to one another (as distinct from antiparallel) and that they lie in axial register [84,90,91,92].

The arrangement of the molecules with respect to one another in the filament has been determined through the specific crosslinking of adjacent lysine residues, using the periodate-cleavable bi-functional cross-linking reagent disulpho-succinimidyl-tartrate (DST). The crosslinked protein can be cleaved with cyanogen bromide and trypsin, and the peptides resolved by high-performance liquid chromatography (HPLC). Through a comparison of the peptide peaks before and after crosslinking, the peaks that have shifted can be recognized. These are reacted with periodate to yield two smaller peptides and both are then sequenced, thereby revealing the origin of each of the two lysine residues that were crosslinked. From a mathematical analysis (least squares procedure) of all of the crosslink data thus obtained the relative positions of the interacting molecules could be deduced. This has been done for epidermal keratins K1/K10 [90] and K5/K14 [91], as well as for trichocyte keratin in the reduced and oxidized states [92,93], thereby replicating the initial formation of the IF in the hair follicle (reduced state) and in the fully-developed hair (oxidized state). The results were far-reaching. Firstly, all of the crosslink data were compatible with three, *and only three,* modes of molecular assembly. These were termed A_11_—which corresponded to the approximate axial overlap of antiparallel 1B segments; A_22_—which corresponded to the approximate axial overlap of antiparallel 2 segments; A_12_—which corresponded to the approximate axial overlap of antiparallel coiled-coil rod domains (Figure 11). Secondly, in the reduced and oxidized states of trichocyte keratin IF the A_11_ values differed by about 2.82 nm whereas the A_12_ and A_22_ values remained unaltered [94]. This implied that a simple axial shift occurred between molecular strands in the filament arising from the change in state from reduced to oxidized [95]. Furthermore, X-ray evidence showed that, at the same time as the axial rearrangement occurred, the filaments underwent a compaction event such that their diameters were reduced by about 1 nm (Figure 12; [81,82,93,96]; see a summary [71,97,98]). Thirdly, after the axial shift has occurred the cysteine residues in different molecular strands in the A_11_ mode were seen to lie in near perfect axial register [99], strongly suggesting that they would form disulphide bonds and thereby endow the final structure with considerable stability (as observed) [100].

Scanning transmission electron microscopy (STEM) has indicated that trichocyte keratin IF contain about 32 chains in cross-section [96]. This would be consistent with the formation of eight protofilaments in the filament and each of these would contain four chains in section, i.e., two molecules. Using the crosslinking data and the modes of association that these infer, together, with X-ray diffraction data, it is possible to find a surface lattice structure for the KIF in the reduced and oxidized states. These are defined by the axial repeat distances, the axial projections of the surface lattice lengths *a* and *b* (*z_a_* and *z_b_*, respectively), and the angular projections of each length (*t_a_* and *t_b_*, respectively) measured around the filament axis. Physically, *z_a_* corresponds to the axial distance between consecutive surface lattice points and *z_b_* to the axial stagger between adjacent protofilaments. The reduced lattice structure is given by an axial repeat of 44.92 nm, *z_a_* = 11.23 nm, *t_a_* = −90°, *z_b_* = 16.85 nm, and *t_b_* = +45° [101]. The oxidized surface lattice, after it has undergone axial rearrangement and lateral compaction, has a helical dislocation in the protofilament packing and is defined by an axial repeat of 47.0 nm, *z_a_* = 7.42 nm, *t_a_* = −90°, *z_b_* = 19.79 nm, and *t_b_* = +45° [102,103,104]. In both cases, however, eight protofilaments are arranged on a ring of constant radius (3.5 nm for the reduced structure and 2.9–3.0 nm for the oxidized one). The protofilaments comprise an antiparallel pair of molecular strands, which assemble via the A_11_ and A_22_ modes previously described. In turn, the protofilaments assemble with each other via the A_12_ mode. Details of the analyses that allowed these lattices to be determined are given in the reviews by Fraser and Parry [71,97,105]. 

In summary, therefore, all of the studies described above are consistent with there being a common structural framework in KIF whether they originate from the sauropsids (tuatara and lizard, but also other members of the sauropsid family) or the vertebrates, such as human, fish, amphibian, and chicken. Thus, the sauropsid keratin molecules will be comprised of Type I/Type II heterodimers with the chains lying in axial register and parallel to one another [84,85,87,88,89,90,91]. In turn, the molecules will assemble to form filaments using the same three modes (A_11_, A_22_, A_12_) characterized experimentally for KIF in mammalian and skin KIF [85,90,91,92]. In addition, the possibility that the sauropsid trichocyte KIF may also have two structures, one pertaining to the initial filament structure produced in a reducing environment and the other to the more compact and axially rearranged structure in an oxidising environment is of interest.

### 3.3. Sauropsids Do Not Contain the Families of Matrix Proteins Seen Around Mammalian Trichocyte KIF 

There is no evidence, however, that any of the three keratin-associated protein (KAP) families associated with mammalian trichocyte KIF (ultra-high sulphur (UHS), high-sulphur (HS) or high-glycine-tyrosine (HGT) families) occur in the sauropsid epidermal appendages [36]. A study of the tuatara partial genome has failed to find any such proteins. There are, of course, many other proteins present that are part of the epidermal differentiation complex in the sauropsid epidermal appendages, but it is not known whether or not these might play a similar role to that of the KAPs in the mammals. One possibility that has been suggested [36] is that the cysteine-rich 3.4 nm diameter filaments formed from the corneous ß-proteins (and which are absent in the mammals) might do so. However, there is no direct biochemical evidence to support this suggestion at the present time and further experimentation will be required to verify or disprove this point. 

## 4. Potential Interactions between Proteins Constituting the Epidermal Appendages

If indeed the cysteine-rich 3.4 nm diameter filaments formed from the corneous ß-proteins in the sauropsids replace the cysteine-rich UHS and HS matrix proteins in the mammals, it follows that the two protein systems would be expected to have closely related roles and functions with respect to the keratin intermediate filaments with which they would interact [36]. It is, therefore, of interest that the framework of the 3.4 nm diameter filaments is net acidic and that the terminal (matrix-forming) domains of these molecules are both apolar and net basic [67]. These same characteristics are present in both the portion of the KIF chains that form the filament framework (net acidic) and in the terminal (matrix-forming) domains (apolar and net basic) [76,98]. Further, in the mammalian keratins the UHS and HS proteins are also apolar and net basic [106]. These observations indicate a common theme of net acidic filaments and apolar and net basic matrix components (terminal domains and matrix proteins) and hence that ionic interactions between oppositely charged residues in the hydrated matrix are unlikely to be either numerous or important. In contrast, the emphasis would likely be on non-specific apolar interactions that are inherently capable of forming a wide variety of energetically-similar packing arrangements. In this situation any spatially adjacent cysteine residues would likely form disulphide bridges on a random basis. This would be consistent with the inherent nature of an unstructured matrix as regards the constituent protein domains and of the various other EDC molecules present. That does not, of course, impinge on the notion that there are almost certainly localized pieces of regular secondary structure present in most constituents of the matrix [106].

Stabilizing covalent bonds between the constituent proteins are not confined to disulphide bridges. Indeed, a host of proteins in the EDC are extensively crosslinked by transglutaminases. These proteins too will also play an important part in determining the overall structure of the matrix, as well as specifying its functional role (see, for example, [33]). Such proteins include loricrin, involucrin, and small proline-rich proteins (all simple epidermal differentiation complex proteins) and S100 fused-type proteins and S100 filaggrin-type proteins [33]. The latter also undergo limited proteolysis and have been shown to interact with other proteins via the S100 domain, as well as through other motifs. Unfortunately, the limited experimental data currently available for the majority of the EDC proteins limits useful speculation at the present time. Clearly, there is still much to learn about these proteins and the interactions that occur between them. This is surely likely to be a profitable focus for future research.

## 5. Summary

Limited structural data on each of the two constituent filaments in the epidermal appendages of sauropsids began to appear in the literature some 90 years ago. It is, of course, in much more recent times that the details of the molecular conformations of the constituent ß-corneous proteins and the α-favouring KIF proteins have emerged at near atomic resolution. In the former case data from fibre X-ray diffraction of seagull feather rachis, polarized infrared spectroscopy, sequence analyses and model building have clearly revealed that a twisted ß-sheet based on a 34-residue sequence conserved across the species assembles with one from a different chain to form an antiparallel right-handed ß-sandwich. On the basis of the X-ray data, the sheets are most likely related to one another by a dyad axis of rotation perpendicular to the fibre axis with line group sr2. Fiber diffraction showed that these sandwiches assemble axially with a left-handed four-fold screw symmetry to form the core of the filament. The terminal domains of the molecules thus constitute the matrix which surrounds each filament. 

Modes of molecular assembly have been characterized for the KIF, albeit from a mammalian keratin homologue rather than one from the sauropsids, as has the surface lattice arrangement of the molecular repeating units in the filament. Additionally, molecular dynamics studies and results from X-ray crystallography have painted an ever-increasingly detailed picture of the KIF structure at atomic resolution. 

Thus, while much of the structures of the conserved portions of the two filamentous molecules are well defined there is now an increasing need to focus on the structures of the terminal domains of these molecules and also on the proteins of the Epidermal Differentiation Complex (EDC). Details of their interactions with one another and also with each of the two filamentous structures represents the next important step in gaining a full appreciation of the physical properties of each of the epidermal appendages in the sauropsids.

## Figures and Tables

**Figure 1 genes-12-00591-f001:**
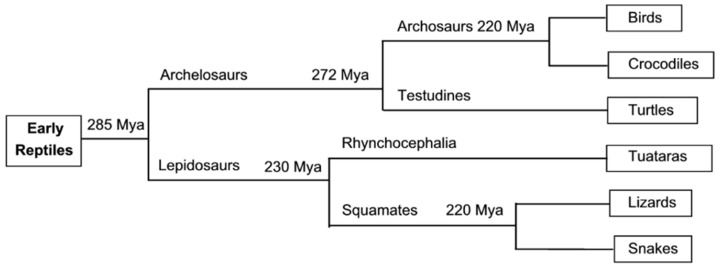
Phylogenetic classification of the sauropsids. The branching times are measured in millions of years (Mya) and are based on studies of mitochondrial DNA [1]. Figure reproduced from [5] with permission of Elsevier.

**Figure 2 genes-12-00591-f002:**
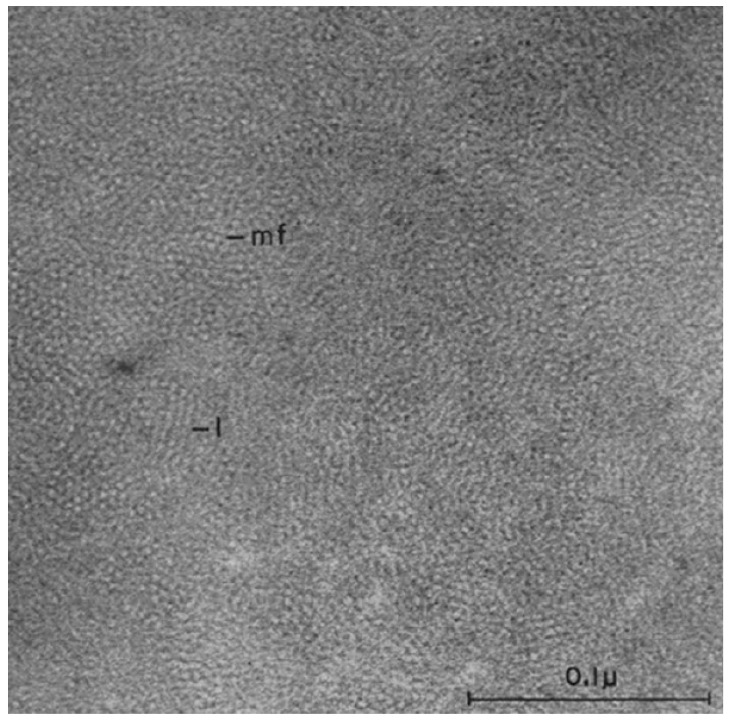
Transmission electron micrograph of a cross-section of feather keratin showing filaments about 3–4 nm in diameter [6]. Figure reproduced from [31] with permission of Elsevier.

**Figure 3 genes-12-00591-f003:**
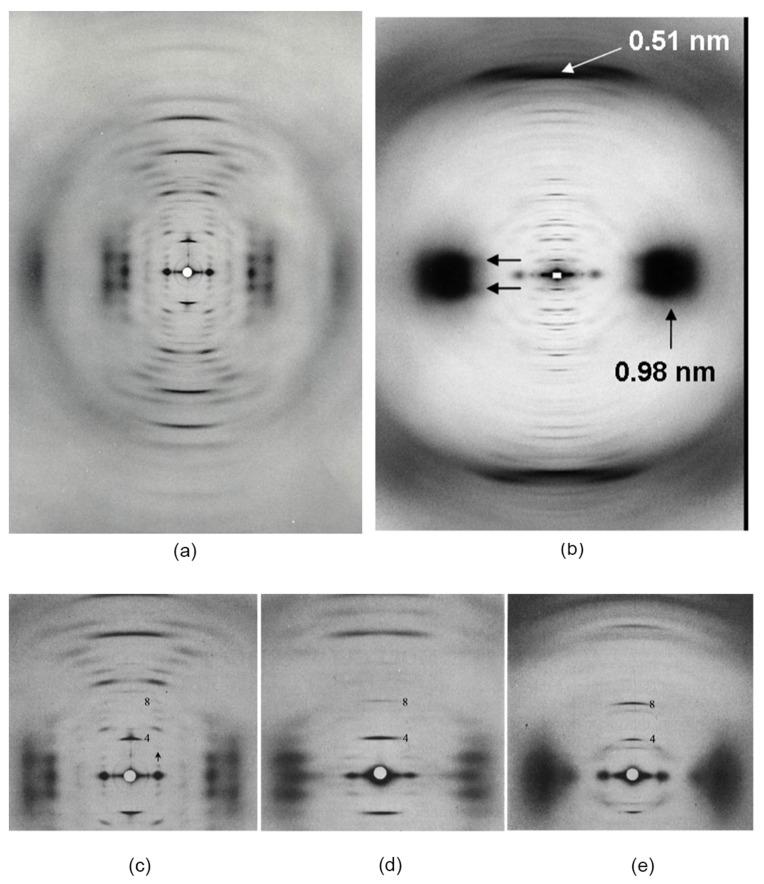
High angle X-ray diffractions patterns of (**a**) seagull feather rachis (*Larus novae-hollandiae*), which shows features of a ß-conformation, and (**b**) porcupine quill (*Hystrix cristata*), which is based on the α-conformation with its characteristic 0.51 nm meridional reflection and the 0.98 nm equatorial and near-equatorial reflections, (**c**,**d**) show the same seagull feather rachis before and after pressing in steam to remove the lateral organisation between the filaments and (**e**) goanna claw (*Varanus varius*). (**b**) reproduced from [43] with permission of Elsevier; (**a**–**e**) reproduced from [44] with permission of Elsevier.

**Figure 4 genes-12-00591-f004:**
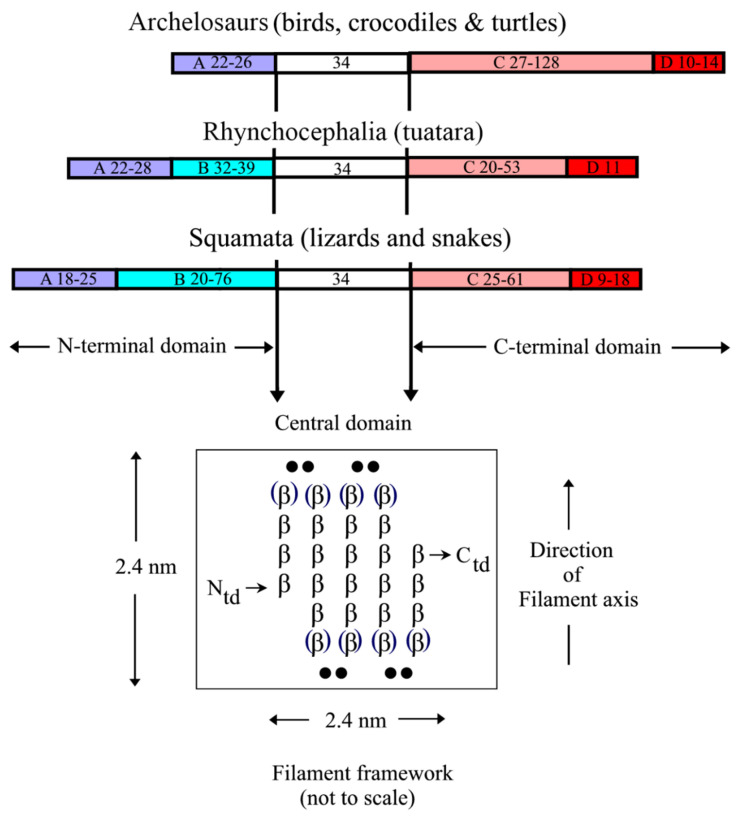
Sequence comparisons of the CBPs from the sauropsids reveal that the N-terminal domain is comprised of subdomain A in the archelosaurs, but subdomains A and B in the squamates and rhynchocephalia. The C-terminal domain, however, comprises subdomains C and D in all of the sauropsids. The sequence characteristics of each subdomain are listed in the text. The approximate size ranges of the subdomains are listed. The central 34-residue domain adopts a twisted ß-sheet structure with three inner strands and two partial outer ones that assembles in an antiparallel manner with a similar sheet from a different chain to form a ß-sandwich. In turn, these assemble axially with a four-fold screw symmetry to form a filamentous structure of diameter 3.4 nm. Figure reproduced from [58] with permission of Springer.

**Figure 5 genes-12-00591-f005:**
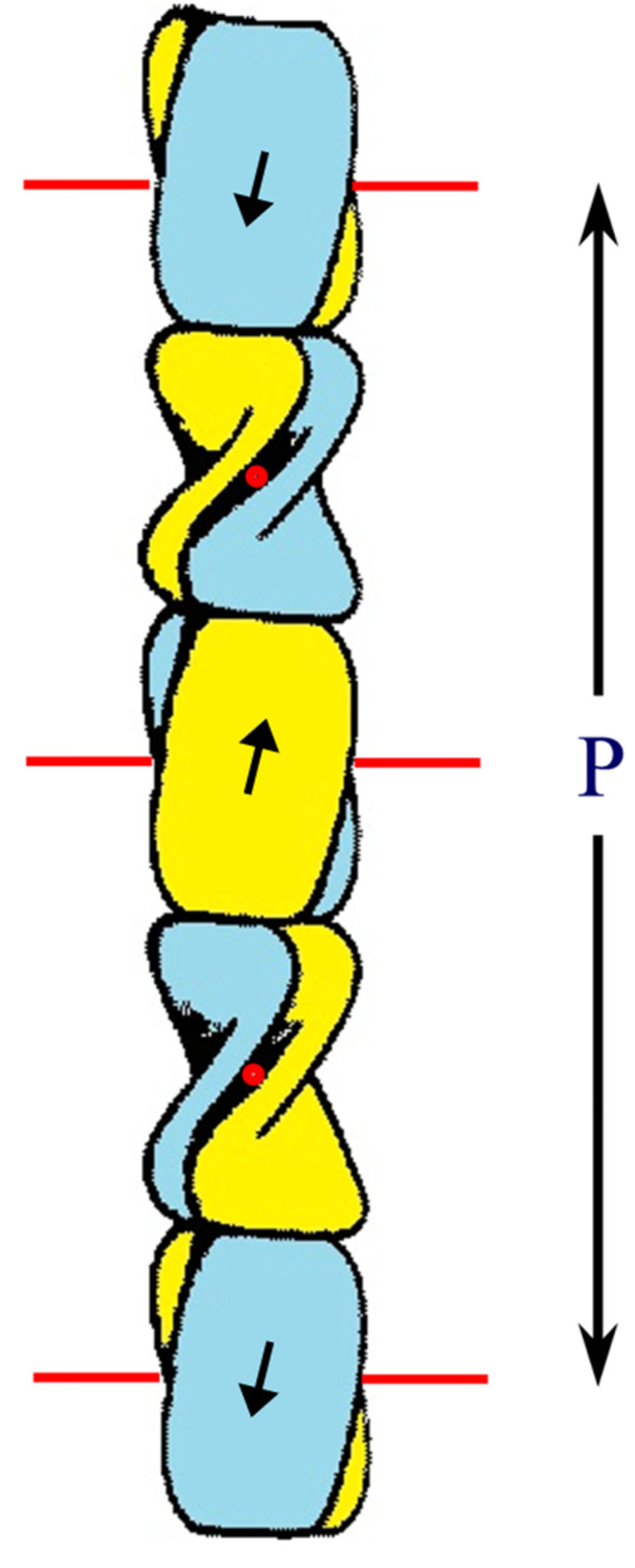
Schematic diagram of the ß-sandwich formed from the conserved 34-residue stretch of sequence in each of two CBP chains. Each sandwich is comprised of a pair of right-handed twisted antiparallel ß-sheets (one is drawn blue and the other is yellow) and these are related to one another by a perpendicular dyad axis of rotation (marked alternatively by red circles and red lines). Axial assembly of the ß-sandwiches through the application of a left-handed four-fold screw axis generates a 3.4 nm diameter filament of pitch length 9.6 nm and axial rise 2.4 nm. Figure reproduced from [58] with permission of Springer.

**Figure 6 genes-12-00591-f006:**
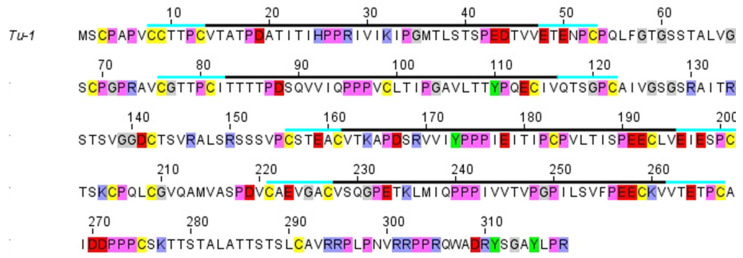
Amino acid sequence of the tuatara chain with four 34-residue repeats. Each repeat is marked with a solid black line. Bars at either end (turquoise) indicate short conserved regions believed to have a role in stabilization or assembly. Color coding for selected amino acids is as follows: cysteine (yellow), proline (magenta), glycine (grey), tyrosine (green), acidic residues (red), and basic residues (blue). Figure reproduced from [5] with permission of Elsevier.

**Figure 7 genes-12-00591-f007:**
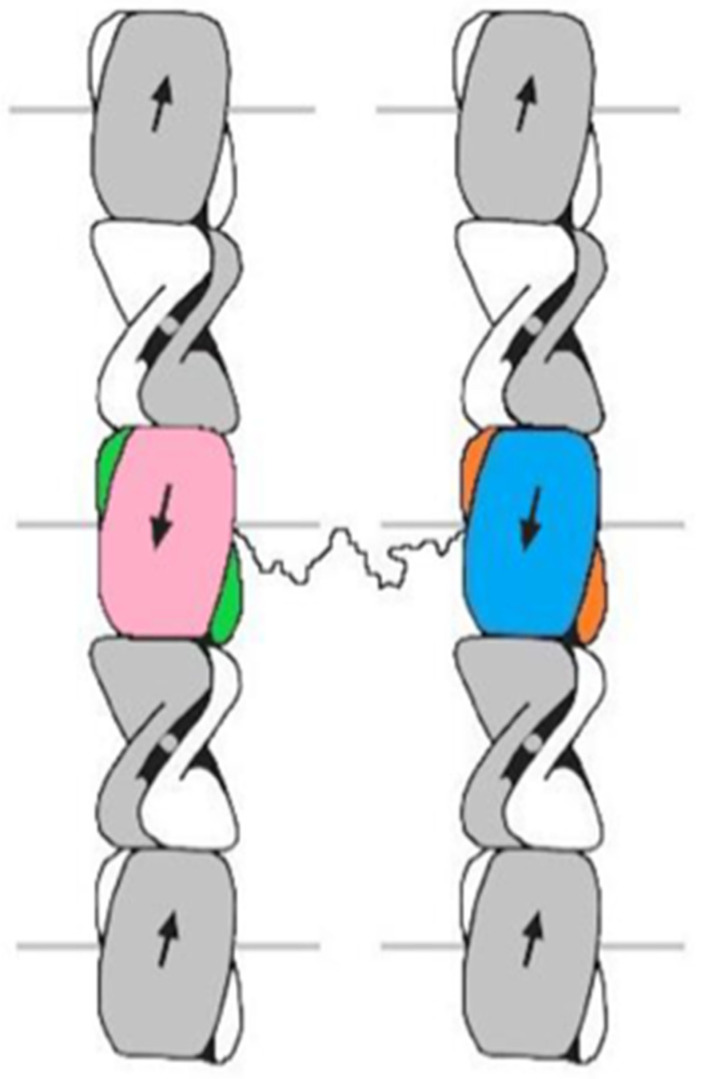
The structure proposed for the chains with four 34-residue repeats contains two ß-sandwiches (pink/green and blue/orange). One occurs in each of two neighboring filaments and are linked by the sequence connecting repeats 2 and 3 (represented by a wavy line). The linker is believed to adopt a ß-like sheet characterized by stretches of sequence with a serine-X repeat. Figure reproduced from [67] with permission of Elsevier.

**Figure 8 genes-12-00591-f008:**
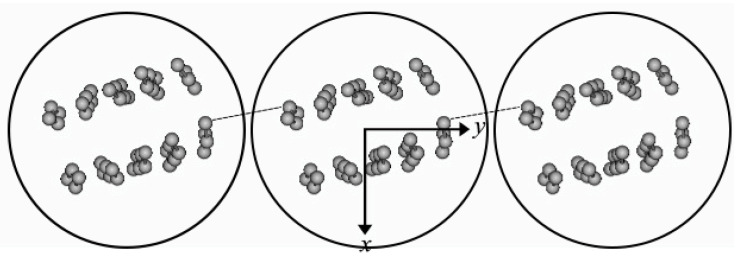
An axial projection of neighboring filaments in feather keratin showing their common orientation, as deduced from the row line pattern seen in the X-ray diffraction pattern. The ß-sandwiches are inclined at an angle of 17° to the plane of the sheet and this results in a ß-sheet in one filament lining up with a ß-sheet in a neighboring filament. Reproduced from [63] with permission of Elsevier.

**Figure 9 genes-12-00591-f009:**
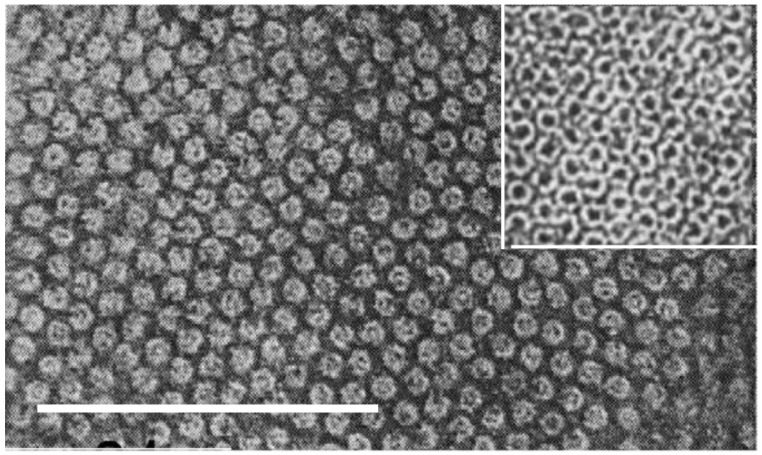
Electron micrograph of a cross-section of fine Merino wool showing the quasi-hexagonal packing of keratin intermediate filaments about 7–10 nm in diameter embedded in an osmiophilic matrix [70]. The magnification bar is 100 nm. The inset (at the same magnification) shows porcupine quill post-stained with potassium permanganate. The KIF have a ring structure. Figure reproduced from [71] with permission of Springer.

**Figure 10 genes-12-00591-f010:**
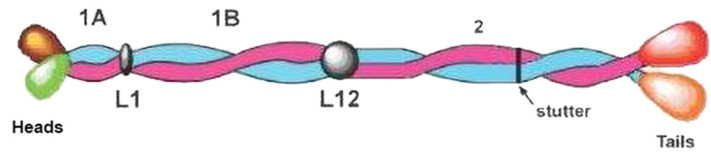
Schematic structure of the KIF molecule with a Type I and a Type II chain in axial register and parallel to one another. The 1A, 1B, and 2 segments have a heptad substructure and form a left-handed two-stranded coiled-coil with connecting linkers L1 and L12. At the N-terminal end of segment 2 the constituent chains have a hendecad repeat which causes the chains to lie approximately parallel to the axis rather than being coiled about it. The N-terminal (head) and C-terminal (tail) domains enclose the central rod domain. Figure reproduced from [71] with permission of Springer.

**Figure 11 genes-12-00591-f011:**
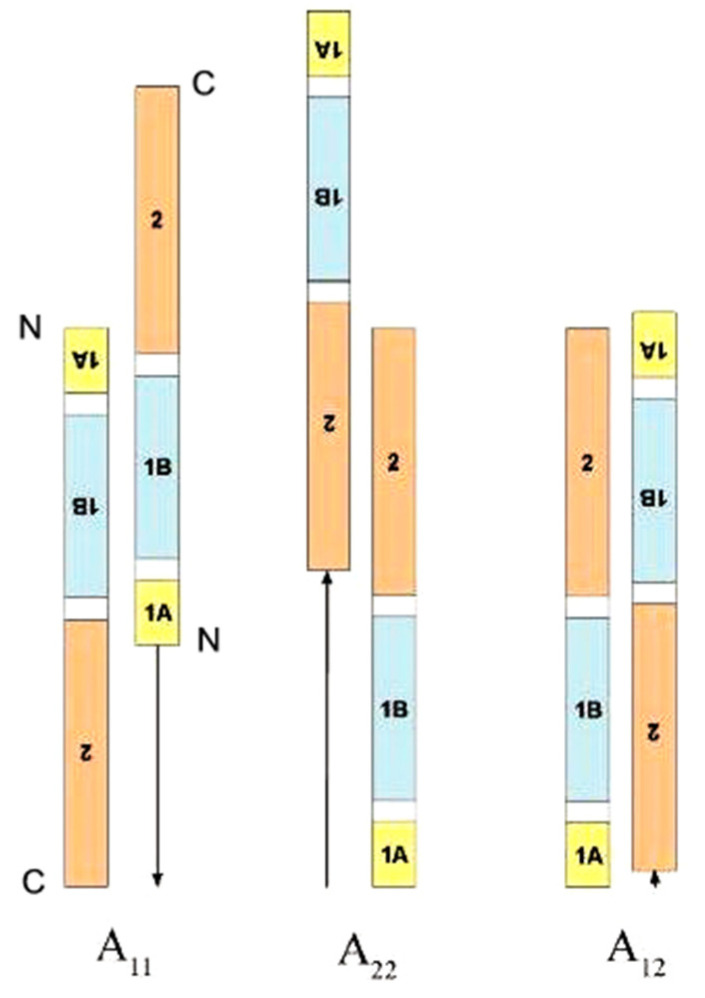
In total, three modes of molecular assembly are found in IF [82,87]. These are termed A_11_, A_22_, and A_12_ and involve the approximate axial alignment of antiparallel 1B segments, antiparallel 2 segments and antiparallel rod domains, respectively. The axial staggers for each mode are measured from the N-terminus of segment 1A in an up-pointing molecule. Figure reproduced from [71] with permission of Elsevier.

**Figure 12 genes-12-00591-f012:**
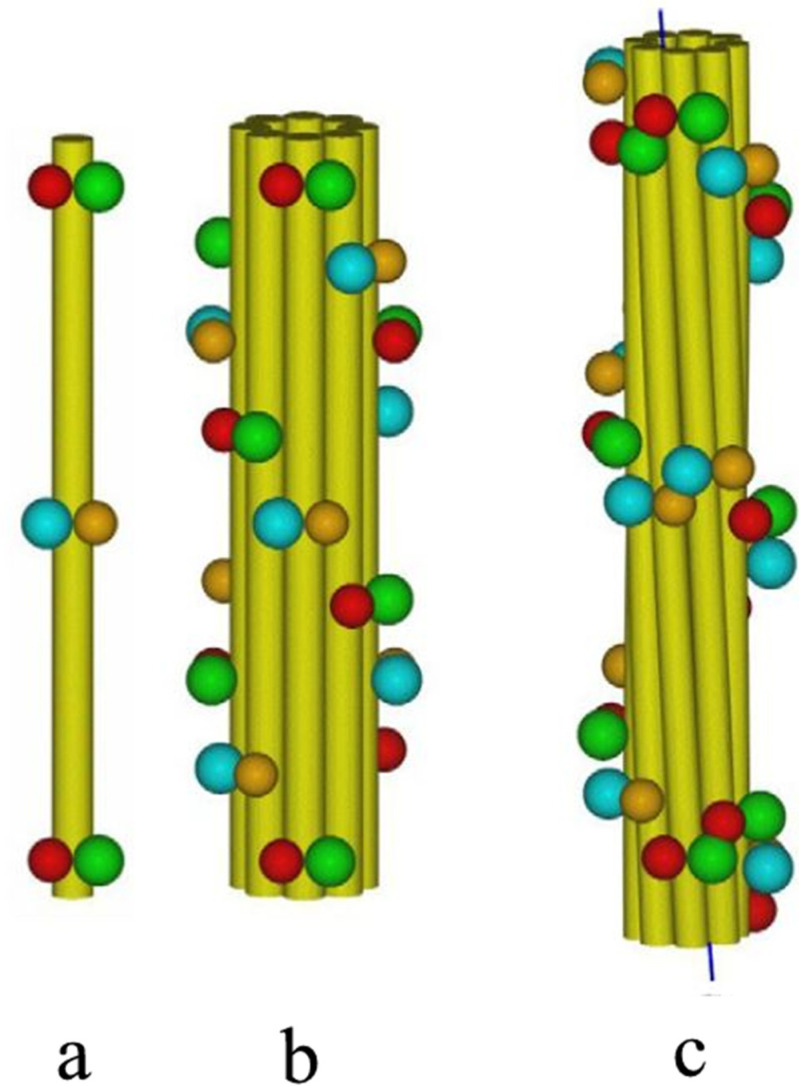
Models of (**a**) a single protofilament in the reduced state, (**b**) the reduced IF with eight protofilaments arranged on a ring of radius 3.5 nm, and (**c**) the oxidized IF, again with eight protofilaments, arranged on a ring of radius 3.0 nm. The yellow cylinders represent the rod domain of the IF molecules, whereas the spheres represent the terminal domains: green = both heads of the Up strands, red = both tails of the Up strands, blue = both heads of the Down strands, and orange = both tails of the Down strands. In (**b**) the terminal domains in the reduced structure are arranged on a two-start left-handed helix, thereby giving rise to a diagonal banding with a spacing of 22 nm. In contrast, in (**c**) the terminal domains in the oxidized structure with its dislocated lattice lie on a one-start helix of pitch length 23.5 nm. Figure reproduced from [97] with permission of Springer.

## Data Availability

No new data were created or analyzed in this study. Data sharing is not applicable to this article.

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
