# Peer review of "Structures of the ß-Keratin Filaments and Keratin Intermediate Filaments in the Epidermal Appendages of Birds and Reptiles (Sauropsids)"

_genes, 2021, doi:10.3390/genes12040591_

Round 1

Reviewer 1 Report

Solid and well-organized structural information is provided for the filament systems found in the epidermal appendages of sauropsids. Perhaps a short section, which would speculate on the possible interactions begin the terminal amino acid sequences of the fiber systems and the conserved amino acid sequences of an EDC, such as one presented by Leopold Eckhard's research group, would be appropriate.   

Author Response

Reviewer #1

Thank you for your helpful comments and, indeed, your very positive review. I understand the point that you have made with regard to the CBP as part of the EDC and I have incorporated the necessary corrections to address them, including addition of the relevant reference. The line numbering in my manuscript differed slightly from yours but I was easily able to locate the places where changes were required.

The two additional references that have been suggested (Greenwold et al and Ehrlich et al) have been added. Thank you for pointing out their omission. I should have included them in my original manuscript as both are important contributions.

Reviewer 2 Report

This is an excellent review of the filament structures in skin appendages of birds and reptiles. I have only few minor comments.

Lines 42-46 and 263-264: It is not completely correct to distinguish between CBPs/beta-keratins as filament components and “the proteins of the EDC”. A paper in 2014 reported “evidence that the sauropsid-specific beta-keratins have evolved as a subclass of EDC genes” (Strasser et al. Mol Biol Evol. 2014 Dec;31(12):3194-205). Therefore, CBPs should also be considered as EDC proteins and the distinction should be between CBPs and OTHER EDC proteins.

Line 332: Please mention reports that have shown sequences of keratins in birds (especially important because the manuscript is submitted to a special issue on Avian Epidermis) and other reptiles in addition to lizard and tuatara. Examples of relevant genomic studies include:

Greenwold MJ, Bao W, Jarvis ED, Hu H, Li C, Gilbert MT, Zhang G, Sawyer RH. Dynamic evolution of the alpha (α) and beta (β) keratins has accompanied integument diversification and the adaptation of birds into novel lifestyles. BMC Evol Biol. 2014 Dec 12;14:249.

Ehrlich F, Lachner J, Hermann M, Tschachler E, Eckhart L. Convergent Evolution of Cysteine-Rich Keratins in Hard Skin Appendages of Terrestrial Vertebrates. Mol Biol Evol. 2020 Apr 1;37(4):982-993.

The letter “ß” should be replaced by β.

Author Response

Reviewer #2

Thank you for your comment. As suggested I have now added a short section speculating on the possible interactions between the various protein components. In particular I have mentioned the potential role of apolar interactions in facilitating the random formation of disulphide bridges as well as the importance of transglutaminase-initiated crosslinks.  Reference to the fine work of Leopold Eckhard and his group has also been included in this context since their reputation and knowledge in this field is clearly superior to my own.